# Wideband Circularly Polarization and High-Gain of a Slot Patch Array Antenna Realized by a Hybrid Metasurface

**DOI:** 10.3390/s24113510

**Published:** 2024-05-29

**Authors:** Qiang Chen, Jun Yang, Changhui He, Di Zhang, Siyu Huang, Min Wang, Fangli Yu, Guanghua Dai

**Affiliations:** Air Force Early Warning Academy, Wuhan 430019, China; yangjem@126.com (J.Y.); hujiabei61@163.com (C.H.); deeafeu@163.com (D.Z.); hsy_19941208@163.com (S.H.); 13995668024@163.com (M.W.); yufangli_aewa@163.com (F.Y.); oumai@163.com (G.D.)

**Keywords:** circularly polarized (CP), hybrid metasurface (HMS), CMA, low profile

## Abstract

In this paper, a patch array antenna with wideband circular polarization and high gain is proposed by utilizing a hybrid metasurface (MS). A corner-cut slotted patch antenna was chosen as the source due to the possible generation of CP mode. The hybrid MS (HMS), consisting of a receiver MS (RMS) arranged in a 2 × 2 array of squared patches and a linear-to-circular polarization conversion (LCPC) MS surrounding it was then utilized as the superstrate driven by the source. The LCPC MS cell is a squared-corner-cut patch with a 45° oblique slot etched, which has the capability for wideband LCPC. The LCPC unit cell possesses wideband PC capabilities, as demonstrated by the surface current analysis and S-parameter simulations conducted using a Floquet–port setup. The LP EM wave radiated by the source antenna was initially received by the RMS, then converted to a CP wave as it passed through the LCPC MS, and ultimately propagated into space. To further enhance the LCPC properties, an improved HMS (IHMS) was then proposed with four cells cut at the corners, based on the original HMS design. To verify this design, both CMA and E-field were utilized to analyze the three MSs, indicating that the IHMS possessed a wideband LCPC capability compared to the other two MSs. The proposed antenna was then arranged in a 2 × 2 array with sequential rotation to further enhance its properties. As demonstrated by the measurements, the array antenna achieved an S11 bandwidth of 60.5%, a 3 dB AR bandwidth of 2.85 GHz, and a peak gain of 15.1 dBic, all while maintaining a low profile of only 0.09λ_0_.

## 1. Introduction

Circularly polarized (CP) antennas play a vital role in wireless communication systems and establishing point-to-point links by effectively addressing challenges related to multi-path interference and polarization mismatches. The increasing need for CP antennas with strong signal reception, wide frequency coverage, and a wide 3-dB axial ratio angle has spurred research into various design methods. Among these approaches is the utilization of MSs which have shown great efficacy in producing and enhancing CP radiation. As a result, numerous antennas incorporating MSs have been created and successfully demonstrated wideband CP capabilities [1,2,3,4,5,6,7,8,9,10,11,12]. However, these uniform MS designs are limited in their ability to effectively enhance bandwidth and improve gain, which results in narrow applicability.

Then, in recent work, an increasing number of non-uniform MSs have been used in the design of CP antennas due to their flexible control of phase correction, resulting in improved radiated performance [13,14,15]. In our previous work [16], we proposed a nonuniform MS design consisting of a corner-cut slotted patch at the center and 45° oblique slotted square patches surrounding it as a radiator, which then achieved significantly improved gain and bandwidth. As reported in [17], a nonuniform MS arranged with unequal patches was demonstrated to achieve a bandwidth greater than 84% and an aperture efficiency of 96%. In particular, in [18], a non-uniform MS was proposed with a centered 2 × 2 array of unequal patches surrounding it, and then arranged in a 2 × 2 array with sequential rotation, ultimately achieving a remarkable enhancement in bandwidth and reduction in RCS. However, the MS-based antenna cell itself cannot achieve CP radiation. An introduction is made of a 2 × 2 nonuniform MS arranged with simple rectangular patches. By employing an appropriate feed network design, orthogonal modes were achieved, leading to remarkable radiated behavior, as reported in [19]. It is worth noting that CP modes are excited through both the feed network and the MSs, which adds complexity to the antenna design. But, this approach confirms that employing MSs as a substrate or superstrate has been a successful method for enhancing the bandwidth of CP patch antennas.

However, enhancements in bandwidth and gain improvements were still limited due to the imprecise control of the wave front. Based on the analysis of the aforementioned works, it can then be found that receive and transmit stages were not separated when utilizing those MSs, leading to phase cancellation at certain frequencies. Based on this consideration, a receiver-transmitter MS (RTMS) was proposed with high aperture efficiency and high gain. The energy is received by a receiver patch and transmitted to the transmitter through a metal via, with the receiver separated from the transmitter by a patch, as reported in [20]. Following this, in the next design, we can separate the receiver and transmitter stages to further enhance the properties through independent phase control.

Then, in this paper, we propose a novel design to achieve CP and high gain by adopting an HMS. As shown in Figure 1, the electromagnetic waves emitted from the driver patch are received by the MS and then pass through another MS, merging into surface waves that can generate additional resonance [21]. In contrast to conventional MSs [22,23,24], this design utilizes an HMS consisting of RMS and LCPC MS, which can optimize radiation behavior. Specifically, the LCPC MS can also achieve a reduction in AR values, resulting in enhanced CP bandwidth and additional resonance. To further enhance the properties, an improved HMS was then proposed with the four patches at corners cut. To explain the design process, both CMA and E-field analysis were utilized to analyze the radiated behavior. A 2 × 2 HMS-based array antenna with sequential rotation was proposed, resulting in significantly enhanced properties. Simulated and measured results then effectively demonstrated the properties of the designed antenna.

## 2. CP Antenna Element Design Process and Operating Mechanism

### 2.1. Souce Antenna Structure

As depicted in Figure 2, a corner-cut slotted patch antenna was used as the initial antenna in this design, namely ele.a, which had been employed in our previous work [16]. The corner-cut slotted patch is connected to the ground plane through an off-centered metal via, potentially achieving a reduction in axial ratios (ARs) through CP design methods. The sandwiched substrate is made of Rogers RT ($\varepsilon_{r}$ = 2.2, tanδ = 0.0014) with a height of 3.175 mm (*h*_a_) and is sized by 55 mm × 55 mm. Further optimized detailed dimensions are listed in Table 1. Ele.a was then simulated in HFSS.15, and the results are shown in Figure 3. As shown in Figure 3a,b, we can observe that a narrow S11 bandwidth and low main-lobe gain were achieved. The minimum AR did not fall below the 3 dB line, indicating that the CP modes were not excited in this design.

### 2.2. Analysis of LCPC Unit Cell and Hybrid MS Design

Hence, to further excite the CP modes in the subsequent design, some additional techniques should be employed. Following this, a polarization conversion MS (PCMS) superstrate is then considered for the unitization of the next CP design. The sketch of the PCMS unit cell, shown in Figure 4a, consists of squared-corner-cut patches with oblique slot etched, and the ground plane separated by the RT–Rogess substrate. To investigate the principle of linear-to-circular polarization conversion, the surface currents on the LCPC unit cell were analyzed, as shown in Figure 4a. Assuming an incident wave with x-polarization along the x-direction (${\vec{E}}_{ix}$), surface currents were excited along the x-direction. However, due to the asymmetric structure, there is a difference in impedance along the x and y directions, leading to the excitation of both x-polarized (${\vec{E}}_{rx}$) and y-polarized waves, as depicted in Figure 4b. The total field (${\vec{E}}_{t}$) can be expressed as ${\vec{E}}_{t}={\vec{E}}_{ix}+{\vec{E}}_{rx}+{\vec{E}}_{ry}$, representing the superposition of the incident field (${\vec{E}}_{ix}$), and the reflected fields (${\vec{E}}_{rx}$ and ${\vec{E}}_{ry}$). Given that the impedance in the x-direction significantly exceeds that in the y-direction, the magnitude of ${\vec{E}}_{ry}$ surpasses that of ${\vec{E}}_{rx}$, resulting in the total field, ${\vec{E}}_{t}$ being almost considered as the sum of the incident field, ${\vec{E}}_{ix}$ and the reflected field, ${\vec{E}}_{ry}$ along the x and y directions, respectively. This implies that the x-polarized incident field can be converted to a y-polarized reflected field. Through asymmetric design, the LCPC MS can achieve a 90° phase difference and equal magnitude between the incident and reflected fields at a specific frequency upon hitting the MS, as it is well-known as an anisotropic homogeneous structure.

Utilizing a Floquet–port setup depicted in Figure 5, we investigate the reflection properties of the unit cell. The unit cell is exemplified by the presence of incident waves in the x-polarization. When an x-polarized wave is incident, *S_xx_* and *S_yx_* serve as indicators of the reflection coefficients for waves reflected in the x-direction and y-direction, respectively. Enhanced polarization conversion is achieved when *S_xx_* is minimized and *S_yx_* is maximized. This results in the suppression of the reflection wave’s x-direction component and its conversion into a y-direction component across a wide frequency range. At 5.3 GHz, S*_yx_* measures around 0.99 on a linear scale with a negative 90-degree phase shift, suggesting that the wave reflected in the y-direction lags 90 degrees behind the incident wave in the x-direction, thereby generating RHCP waves at 5.3 GHz.

But, following our novel design that separates the receiver and transmit stages, as depicted in Figure 6a, the proposed PCMS superstrate consisted of squared patches arranged in the center, with LCPC unit cells surrounding it, which can then be viewed as a hybrid MS (HMS). According to the functions they fulfill, the HMS can also be regarded as a combination of a receiver MS (RMS) and a PCMS, as detailed in Figure 6. The linearly polarized waves emitted from Ele.a were first received by the RMS and then converted into circular polarization as they passed through the PCMS, ultimately radiating into space as CP waves.

Then, the RMS-only superstrate-based (RMSSA) and HMS superstrate-based antenna (HMSSA) were both simulated in HFSS 15, where the results are plotted in Figure 7. As shown in the Figure 7a, HMSSA achieves an S11 bandwidth of 3 GHz, ranging from 4.1 GHz to 7.1 GHz, while a narrow bandwidth is realized by the RMSSA, with only one resonance frequency. As for the ARs achieved by HMSSA, the bandwidth is wide, at 1.2 GHz, ranging from 4.6 GHz to 5.8 GHz, with the minimum value reaching 1 dB, and the RMSSA could also realize CP radiation, but with a slightly narrower bandwidth of 0.1 GHz, as shown in Figure 7b, demonstrating that the RMS can receive most LP waves ranging in an wide band and reduce AR values in an certain degree. It can also be observed that the HMSSA obtains an average value 1 dB higher than the RMSSA. Through the simulations analysis above, it can then be demonstrated that the hybrid design for MS has the potential to be applied to the wideband CP and high-gain antennas. Moreover, to well validate this design approach further, a squared conventional uniform MS-based superstrate antenna (CMSA) was then also simulated for comparison with the HMSSA, and the results are shown in Figure 7. It can be observed that the HMSSA achieves a wider CP bandwidth, with an additional AR minimum, compared to those of the CMSA, although the similar as in the S11. However, the AR minimums of the CMSA fail to meet the 3 dB line at lower frequencies, and the gains significantly decrease at frequencies higher than 5.5 GHz, as illustrated in Figure 7. Thus, the proposed HMS has a capable wideband PC ability that can be applied to wideband CP and high-gain radiation design.

To further enhance the CP bandwidth and reduce AR values, an improved HMS was then proposed, as depicted in Figure 8. It can be observed that four PCMS units of the HMS at the corner were cut. As shown in Figure 9, the improved HMS-based superstrate antenna (IHMSSA) exhibits an AR bandwidth of 1.8 GHz, ranging from 4.4 GHz to 6.2 GHz, thereby representing an increase of 0.6 GHz compared to the HMSSA. It can be further observed that an average reduction of 1 dB was achieved compared to the HMSSA, with the AR minimum value nearly zero, it confirms a significant enhancement in CP properties. Furthermore, as shown in Figure 9a, there was nearly no influence on S11 after the corner-cut on the HMS, thereby forming IHMS. During the design process, certain parameters play a vital role in achieving better properties, such as the sandwich height, *h_m_*. The simulated results of radiated performance, including S11, AR, and gain, versus frequencies at different heights, *h_m_*, were investigated. The results are shown in Figure 10. It can be observed that the various heights have almost no influence on the S11 bandwidth and peak gain, while significantly affecting AR. Subsequently, the optimal value was determined to be 2.5 mm. The optimal dimensions are listed in Table 1 through numerical simulations.

### 2.3. LHCP IHMS–Antenna Design and Analysis

Additionally, the polarization state can be easily switched by rotating the squared-corner-cut, oblique slot, and corner-cut of the source slot patch all by 90°, which is shown in Section 5, allowing for the realization of an LHCP array antenna. As explained in Section 2, CP modes can be generated when there is a difference in impedance along the *x* and *y* directions, leading to variations in the field of ${\vec{E}}_{rx}$ and ${\vec{E}}_{ry}$. Consequently, the *x*-polarization field can be converted to *y*-polarization. However, it is important to note that the phase lag or lead lag along the *x* and *y* directions results in different polarizations. Therefore, a 90° lag in impedance along the *x* and *y* directions leads to an LHCP state, while otherwise resulting in an RHCP state. It can be concluded that when the squared-corner-cut and oblique slot are rotated by 90°, the phase difference will vary, as expressed by Equation (1), thereby affecting the conversion results.
(1)$$Z=\sum R_{i}+\sum j\omega L_{i}+\sum \frac{1}{j\omega C_{i}}=R^{\prime }+jX^{\prime }.$$

As shown in Figure 11, the LHCP HMS-based antenna achieves an S11 bandwidth ranging from 4.25 to 8.48 GHz by rotating the HMS and corner-cut patch by 90°, resulting in a deeper resonant frequency lower than −40 dB. Additionally, the LHCP antenna exhibits an AR bandwidth of 4.62–5.46 GHz with a minimum lower than 1 dB. The gain remains flat, with a peak gain of 9.4 dBic. On the other hand, as depicted in Figure 12, the Radiation Pattern at 5.3 GHz further demonstrates the polarization state, showing that the LHCP antenna achieves a cross-polarization lower than 30 dB. Therefore, it can be concluded that the CP state can be changed simply through a 90° rotation, as shown in Figure 13. As shown in Table 2, a comparison of the radiated performances with the proposed RHCP IHMS-based antenna was listed. It can be observed that a simple rotation adjustment for the polarization state switch has little to no impact on the radiated performance, validating this design effectively.

## 3. Antennas Analysis

### 3.1. Analysis of the Mode Behaviors of the Two HMSA-Based Antenna Using CMA

To further validate the PC properties of HMS, CMA [16,17,18,19] was then utilized to clearly explain the design process, considering the inherent properties of the conductor structure calculated without the addition of excitation. To achieve circular polarization (CP) properties with CMA-aided design, the following conditions must be met: Firstly, ensure that the two MS values are equal and exceed 0.707; Secondly, maintain a 90° phase difference between the CAs of the two modes; Thirdly, confirm that the current surfaces generated by the two modes are orthogonal; Lastly, ensure consistency in the direction of maximum radiation. The MS values, CAs, surface currents, and radiation patterns were then analyzed using the multilayer solver in CST2019. As shown in Figure 14a, the MSs of the first two modes, with values equal to 1, were observed to occur at 5.05 and 5.93 GHz for RMS, and at 4.78 and 5.89 GHz for the HMS, respectively, which implies that the two MSs resonate at the similar frequencies. Furthermore, resonant frequencies of modes 1 and 5 for both RMS and PCMS are approximately the same as those of the HMSA and IHMSA, as presented in Figure 14a It can be concluded that the modal behaviors of PCMS can be effectively integrated with those of RMS. Then, it thus demonstrates that further design can proceed by combining the two MSs together. From Figure 14b, it can be observed that the phase difference between modes 1 and 2 is 75–105° within the frequency range of 4.8 to 5.8 GHz, indicating that they can be considered as orthogonal modes.

As shown in Figure 15, maximum currents of mode 1 were observed pointing down to the left, while for mode 2, they were in an orthogonal direction, pointing up to the left at 5.5 GHz. Moreover, the current distributions concentrate on the patches in the upper left and lower right positions for mode 1, while on orthogonal positions for mode 2. Upon further investigation in Figure 16, it was observed that the two modes possess similar radiation patterns at 5.5 GHz, mainly directed along the *z*-axis. It is thus concluded that linear-to-circular polarization conversion can be achieved over a wideband by generating modes 1 and 2 in PCMS. That is to say, PCMS can be employed in this design for wideband CP radiation.

As illustrated in Figure 17, the first six modal behaviors of HMS were also explored. As shown in Figure 14a, HMS resonates at 5.56, 6.24, 6.59, 6.74, 4.7, and 6.93 GHz, corresponding to values that are equal to 1. It is thus confirmed that mode 5 should be chosen as the one operation mode because its resonant frequencies fall within the operated band. Through further observation, it was found that only the MS corresponding to the overlapping frequency of modes 1 and 5 is greater than 0.707. Hence, Mode 1 and Mode 5 are chosen as the operation modes for the further CP design. Figure 17b shows the CAs of modes 1–6, where a nearly 90° phase difference between mode 1 and mode 5 at 5.5 GHz can be observed. Currents were also investigated, as shown in Figure 17c. From the figure, it can be observed that the maximum current appears in the lower right direction for mode 1. Mode 2 displays a pair of opposite maximum currents separated on both sides of the diagonal, while the maximum currents for modes 4–6 all point in the lower left direction. As shown in Figure 17d, modes 1 and 5 exhibit a similar radiation pattern along the *z*-axis. We can hence conclude that modes 1 and 5 are a pair of orthogonal modes that can generate CP properties.

Furthermore, the modal behaviors of the IHMS were also analyzed in the design process to achieve enhanced properties. The MSs of the first six modes were depicted in Figure 18b, from which we can see that the operation modes were modes 1 and 4, as the MSs of the overlapped frequency covered by both modes are greater than 0.707 within the operation band. As depicted in Figure 18a, the frequencies band corresponding to the CAs differences ranging from 75° to 105° is wider than that of HMS, with both the lower and higher bands increased. Therefore, it can be indicated that the IHMS can achieve an enhanced CP band at lower and higher frequencies compared to the HMS. It can then be concluded that improved properties were achieved through the adoption of IHMS in this design.

### 3.2. E-Field Analysis for the Three MS-Based Antennas

To further demonstrate this design, we also simulated the E-field radiated from the patches arranged in an MS configuration. It can then be resolved into two mutually orthogonal components, namely Ex and Ey, along the x and y directions. Generally, when the far-field Ex and Ey have equal magnitudes but a 90° phase difference, perfect circularly polarized radiation can be achieved. As shown in Figure 19, the magnitude ratios and phase differences of the two components, Ex and Ey, in the boresight radiated by the three MS have been investigated. It is worth noting that in engineering, CP modes would be generated only when the absolute value of the ratio is not higher than 3 dB and the phase difference variation is not higher than 15° compared to a perfect CP mode. Considering this, it can be observed from the figures that the CP mode generated bands highlighted in blue range from 5.02 to 5.26 GHz for RMS, 4.75 to 5.65 GHz for HMS, and 4.62 to 6 GHz for IHMS, respectively, which is consistent with the results analyzed by CMA. Furthermore, The LCPC frequencies shifted to lower and higher bands, resulting in a wider LCPC band when IHMS was utilized compared to HMS. It can hence be concluded that IHMS has a wider LCPC band than RMS and HMS, which could be applied as a superstrate in this design for wideband CP and high gain.

## 4. Radiating Mechanism of the IHMSA

In order to comprehend how the IHMSA radiates, Figure 20a,b illustrate the simulated E-field distribution on the vertical plane at the center at frequencies of 5.2 and 6.2 GHz. The dispersion diagram of the RMS and HMS unit cells, derived from full-wave simulation, are depicted in Figure 21b, where it is observed that their curves coincide. Therefore, the calculation of the resonant frequencies for the hybrid MS can be then carried out using a single MS unit cell’s dispersion diagram. The frequencies of 5.2 and 6.2 GHz correspond to the first and second resonant frequencies of the IHMSA as depicted in Figure 9a. Furthermore, the anticipated E-field distribution of the TM10 and anti-phase TM20 modes based on the cavity model for a traditional rectangular microstrip antenna is also presented in those figures for comparison. Notably, the E-field distributions at the two resonant frequencies of the proposed IHMS antenna closely resemble the TM10 and TM20 modes of a standard patch antenna, with the exception of the radiation emanating from the gaps between IHMS cell arrays. These radiating gaps contribute to a decrease in the quality factor in comparison to a complete rectangular patch antenna, thereby enhancing the antenna impedance bandwidth.

An additional resonance arises from surface waves traveling along the top layer of the IHMS structure, which proves beneficial for broadening the impedance bandwidth. When the height of the HMSA substrate, hs, is significantly smaller than the wavelength in a vacuum and the width of the HMS array, the resonances can be understood using a simplified transmission line model. The parameters *p_x_* and *p_y_* denote the unit cell period in the *x* and *y* directions, respectively. The resonant frequencies corresponding to the TM10 and TM20 modes can be mathematically represented as specified in reference [25].
(2)$$\beta_{mr}pN_{x}+2\beta_{eff}\Delta L=\pi ,$$
(3)$$\beta_{mr}pN_{x}/2+2\beta_{eff}\Delta L=\pi .$$

The propagation constant, denoted as β, and the quantities N_x_ and N_y_, which signify the count of unit cells in the x and y directions (set at 4 for both in this study), are integral to the behavior of the HMSA array. Due to the presence of fringing fields at the exposed edges of the unit cell array, an additional length extension ΔLx (y) occurs in both the x and y directions. The calculation of this extended length ΔLx (y) in the two directions is detailed in Equations (4) to (7).
(4)$$\frac{\Delta L}{h}=0.412\frac{\left(\varepsilon_{reff}+0.3\right)\left(L_{p}/h+0.262\right)}{\left(\varepsilon_{reff}-0.258\right)\left(L_{p}/h+0.813\right)},$$
(5)$$\varepsilon_{reff}=\frac{\varepsilon_{r}+1}{2}+\frac{\varepsilon_{r}-1}{2}{\left(1+12\frac{h}{W_{p}}\right)}^{-1/2},$$
(6)$$W_{p}=N_{y}p-g.$$

The height (hs) and permittivity ($\varepsilon_{r}$) of the HMSA substrate, along with the effective width (Lp) of the unit cell array in the x and y directions, are defining factors. Equation (6) specifies the propagation constants for the extended regions in both the x and y directions.
(7)$$\beta_{reff}=k_{0}\sqrt{\varepsilon_{reff}}=\frac{2\pi f}{c}\sqrt{\varepsilon_{reff}}.$$

Figure 21b displays the dispersion plots obtained from the S-parameter data of the simulation model. This study employs TM modes within the RH region for the antenna design. The resonant frequencies for the TM10 and counter-phase TM20 modes, calculated using Equations (2) and (3), are 5.26 GHz and 6.35 GHz, respectively. The resonance frequencies for the TM10 and TM20 modes are determined at the intersection points of the dispersion curves, with corresponding values of $\beta_{\mathrm{mr}}P/\pi$ at 0.033 and 0.85. It is evident that the projected resonances for the TM10 and TM20 modes (around 5.26 GHz and 6.35 GHz) closely align with the simulated first and second resonant frequencies (5.2 GHz and 6.2 GHz) of the HMSA antenna. In conclusion, the analysis of dispersion presented above effectively showcases that the radiation mechanism of the proposed antenna can be clearly explained by the transmission-line model.

## 5. Simulated and Measured Results

To further enhance the radiated properties and achieve RCS reduction, a sequentially rotated array arranged in a 2 × 2 configuration was proposed, with the structure then shown in Figure 18. By observing Figure 22a, a feeding network was designed to generate four signals with equal magnitude but with a 90° phase difference from one port, and it was printed on the Rogers RT substrate, namely Sub#3. Figure 22b shows the detailed configuration of the 2 × 2 arranged IHMS array with the gap space, g, and the photographs of the fabricated array antenna are depicted in Figure 22c. Furthermore, the behavior of the network was investigated, as shown in Figure 23. It was observed that the magnitudes of the four ports can be almost the same in a large frequency range of 4–8 GHz, as depicted by the overlapping curves of S12, S13, S14, and S15 in Figure 23a. Additionally, in Figure 23b, the phase difference in the sequence of ports 1, 2, 3, and 4 was maintained at about 90° within the same frequency range of 4–8 GHz. It can thus be concluded that the feeding network can satisfy the properties of equal magnitude but with a 90° phased difference in sequence for the four feeding points. To experimentally verify this design, the proposed IHMSA-based array antenna was then measured in the anechoic chamber, and S parameters were analyzed using a vector network analyzer.

As shown in Figure 24a, the measured S11 is below −10 dB in the range from 4.15 GHz to 7.75 GHz, with a bandwidth of 3.6 GHz. It closely matches the simulations, except for some frequencies that deviate slightly but are still acceptable. Additionally, it can be observed from Figure 24b that the patch array antenna achieved a measured AR bandwidth of 2.85 GHz, ranging from 4.15 GHz to 7 GHz, within the impedance band. As further illustrated in Figure 24b, the main-lobe gain can reach a maximum of 15.1 dB at 5 GHz. While it decreases rapidly at higher frequencies, which can be attributed to challenges in beamforming at those frequencies, the gain value above 12 dB falls within the range of 4–6.35 GHz, illustrating a significant property in this design.

On the other hand, the radiation patterns at 4.7 GHz in the *xoz* and *yoz* planes were also investigated, as shown in Figure 25. It can be observed that the array antenna achieves cross-polarization of less than −35 dB in both the *xoz* and *yoz* planes, with the side-lobe level kept 15 dB lower than that of the main lobe. It is worth noting that the polarization state of the simulation and measurement, RHCP, is consistent with the simulated results of the LCPC unit cell using the Floquet–port setup, which effectively validates our design. Moreover, as shown in Figure 26, the antenna with ARs is lower than 3 dB and has a beamwidth of 60°, ranging from −30° to 30° at 4.7 GHz in the *xoz* plane and 57° in the *yoz* plane with a range of −30–27°, indicating a significant CP property. Furthermore, a comparison with recent related works is listed in Table 3, confirming the superior properties achieved in this design.

## 6. Conclusions

In this paper, we propose an IHMS-based patch array antenna with wideband CP and high-gain properties. The IHMS consists of a 2 × 2 arrangement of RMS and the LCPC MS, which are square-corner-cut slotted patches surrounding it. By cutting four MS cells at the corners, the LCPC properties can be further enhanced with the formation of IHMS. Initially, EM is received by RMS, then converted to CP waves, and finally radiated into space, thereby widening the PC band and increasing the radiated gain through hybrid design. To validate this design, we utilized CMA to analyze the three MSs. The results from CMA indicate that the IHMS has a wider PC band as it satisfies a wider frequency range of MSs and CAs compared to other MSs. Additionally, we explored the E-field radiated by IHMS patches and found that the IHMS achieves a wider PC band while maintaining a magnitude no more than 3 dB and a phase difference variation no greater than 15°. To further enhance the CP properties, we then implemented an array antenna arranged in a 2 × 2 sequentially rotated IHMS-based configuration. Measured results demonstrate that this design achieves wideband CP and high gain with a low profile. Therefore, the proposed patch array antenna has the potential to be applied in point-to-point links and communication systems.

## Figures and Tables

**Figure 1 sensors-24-03510-f001:**
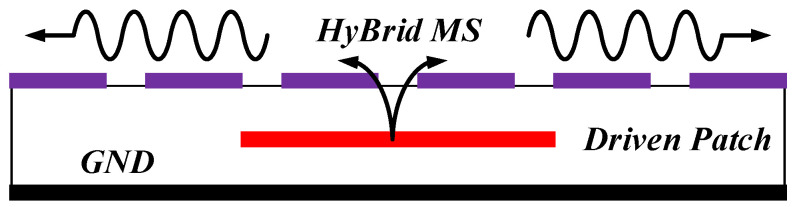
The schematic of the electromagnetic wave propagating on the proposed antenna.

**Figure 2 sensors-24-03510-f002:**
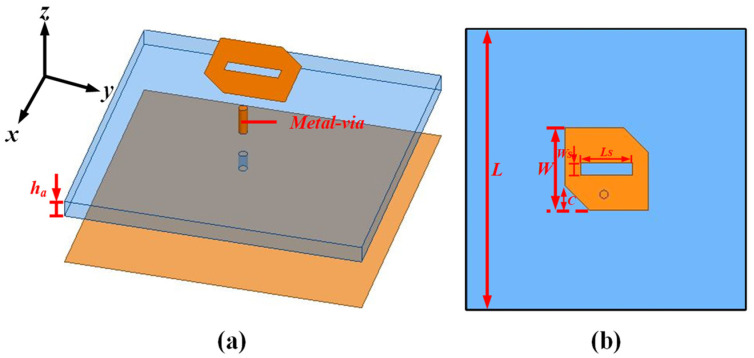
Initial antenna configuration: Element A—(**a**) Perspective view in 3D and (**b**) bottom view.

**Figure 3 sensors-24-03510-f003:**
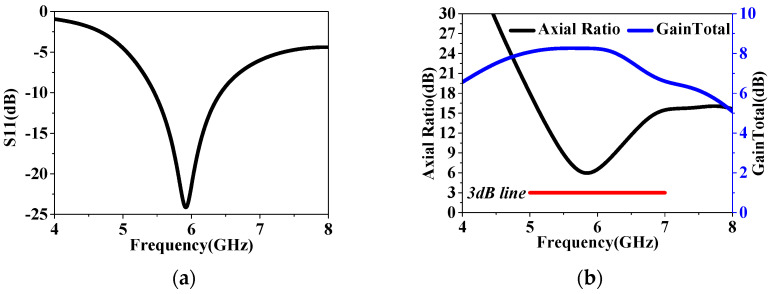
The simulation of the patch–only antenna, namely ele.a: (**a**) S11 and (**b**) AR and GainTotal.

**Figure 4 sensors-24-03510-f004:**
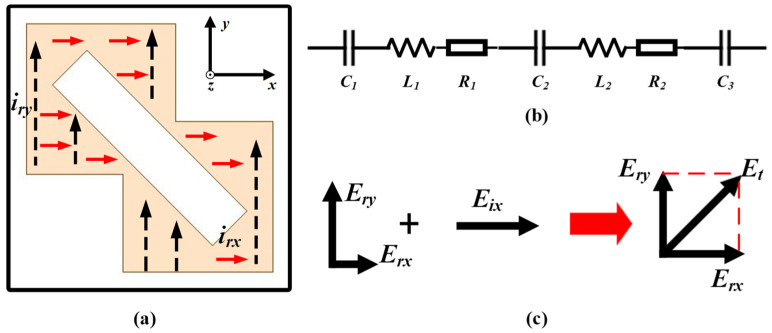
LCPC unit cell: (**a**) surface currents behavior, (**b**) equivalent circuit, and (**c**) the sketch of the superposition of the incident and reflect field.

**Figure 5 sensors-24-03510-f005:**
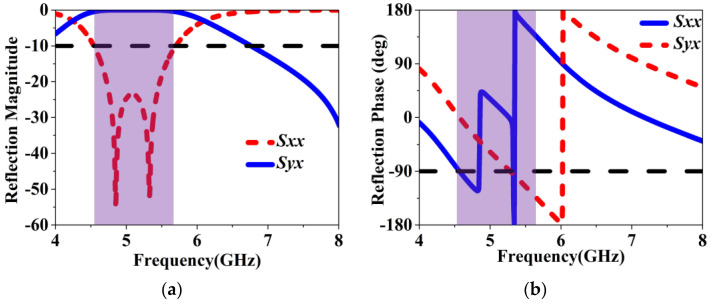
Reflection coefficients of the MS unit under x–polarized wave incidence: (**a**) magnitude and (**b**) phase.

**Figure 6 sensors-24-03510-f006:**
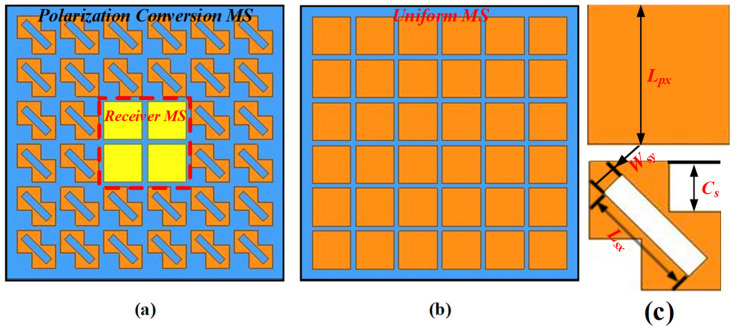
The configuration of (**a**) hybrid MS, (**b**) conventional MS, and (**c**) detailed dimensions of LCPC and RMS cell.

**Figure 7 sensors-24-03510-f007:**
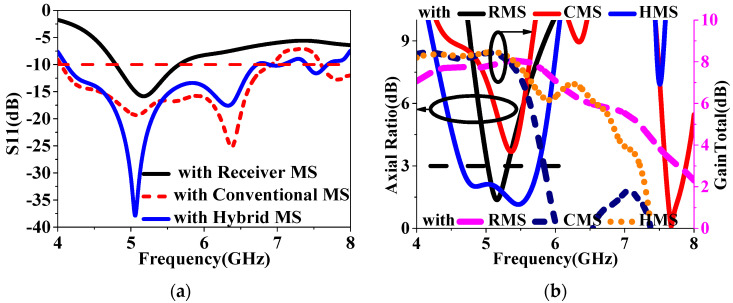
Comparisons of the slotted patch antenna with the three MSs superstrate: (**a**) S11 and (**b**) AR and GainTotal.

**Figure 8 sensors-24-03510-f008:**
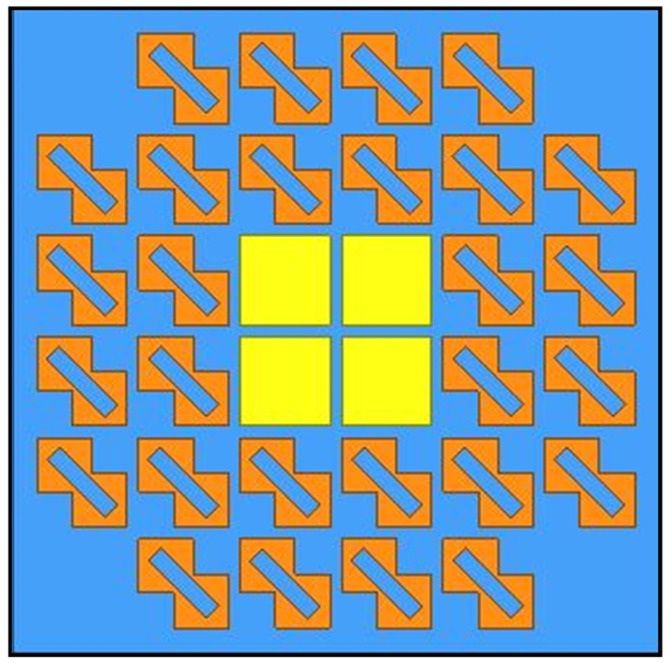
The configuration of the improved hybrid MS.

**Figure 9 sensors-24-03510-f009:**
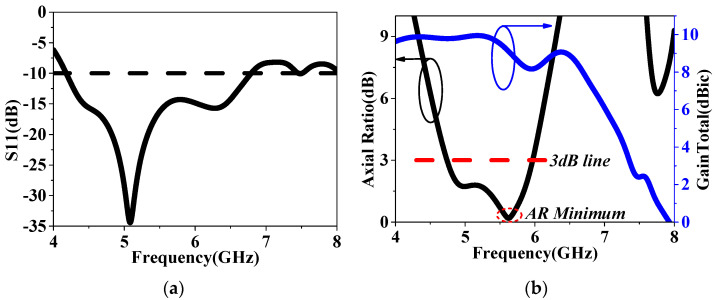
The simulated results of the slotted patch antenna with the improved HMS superstrate: (**a**) S11 and (**b**) AR and GainTotal.

**Figure 10 sensors-24-03510-f010:**
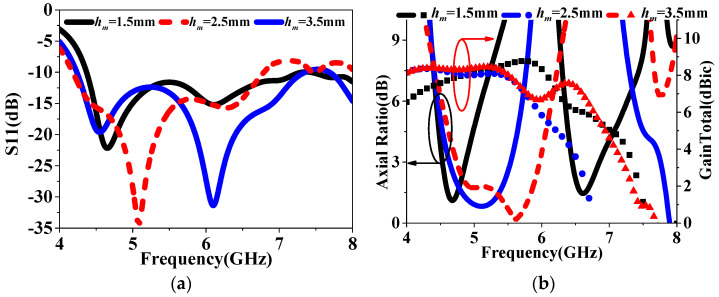
The simulated results of (**a**) S11 and (**b**) Axial Ratio and GainTotal versus frequencies with different height, *h_m_*.

**Figure 11 sensors-24-03510-f011:**
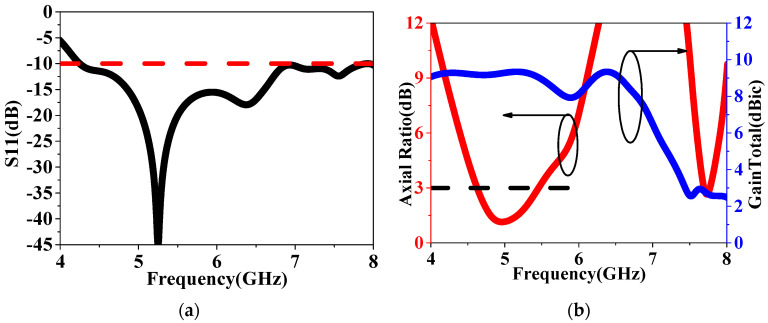
Simulated results of the LHCP HSMS–based antenna: (**a**) S11 and (**b**) AR.

**Figure 12 sensors-24-03510-f012:**
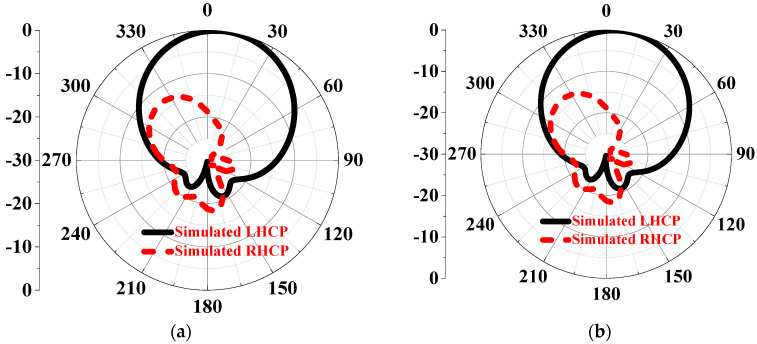
Simulation of the radiation pattern of the LHCP HMS–based antenna at 5.3 GHz: (**a**) *xoz* plane and (**b**) *yoz* plane.

**Figure 13 sensors-24-03510-f013:**
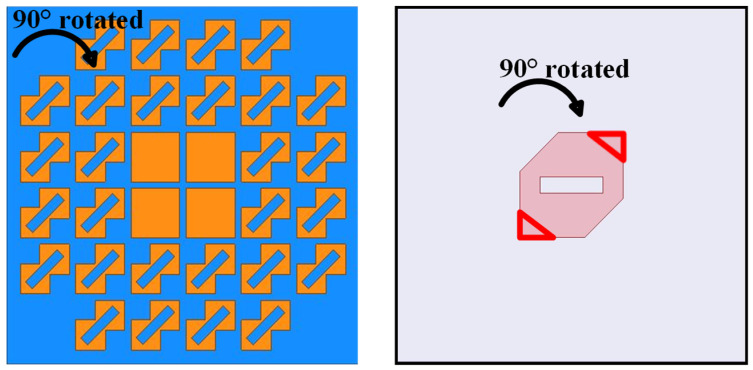
The configuration of the LHCP hybrid MS-based antenna.

**Figure 14 sensors-24-03510-f014:**
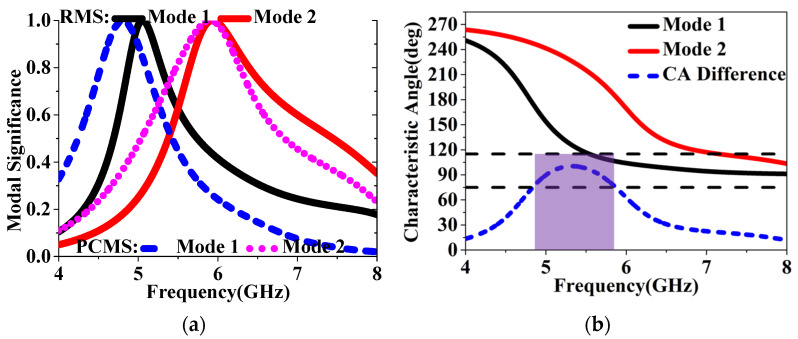
(**a**) MS values and (**b**) CAs of mode 1 and mode 2 of RMS and PCMS, and the difference between them.

**Figure 15 sensors-24-03510-f015:**
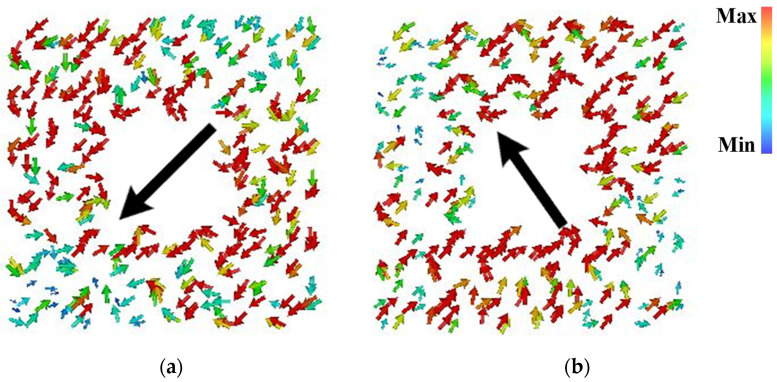
Surface currents of (**a**) mode 1 and (**b**) mode 2 at 5.5 GHz.

**Figure 16 sensors-24-03510-f016:**
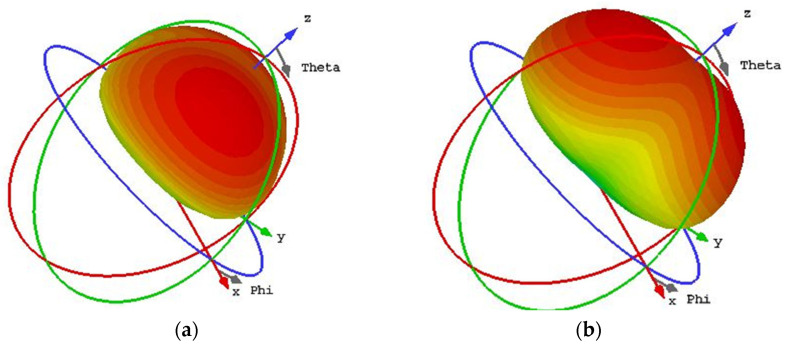
Radiation patterns of (**a**) mode 1 and (**b**) mode 2 at 5.5 GHz.

**Figure 17 sensors-24-03510-f017:**
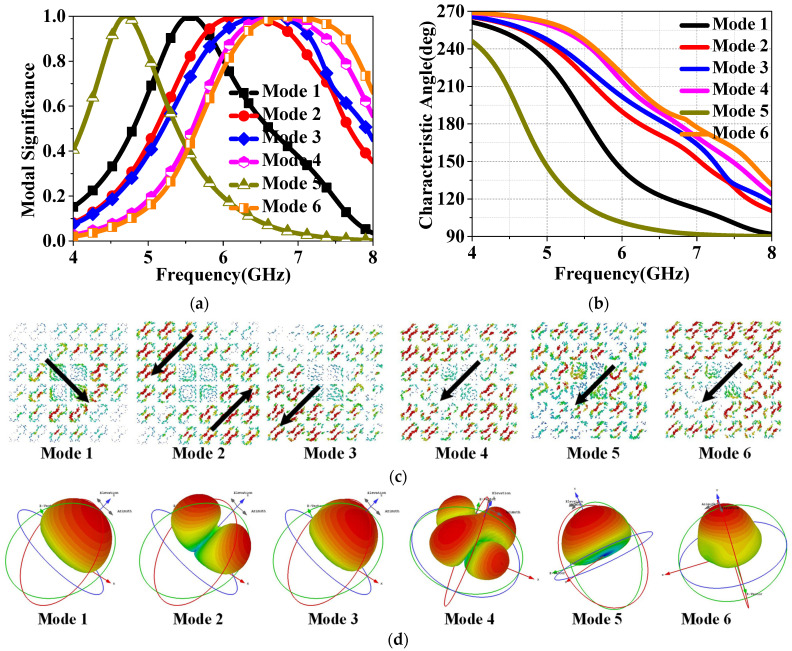
Modal behaviors of the proposed HMSA: (**a**) MS values; (**b**) CAs; (**c**) Surface Currents; (**d**) Radiation Pattern at 5.5 GHz of the first six modes.

**Figure 18 sensors-24-03510-f018:**
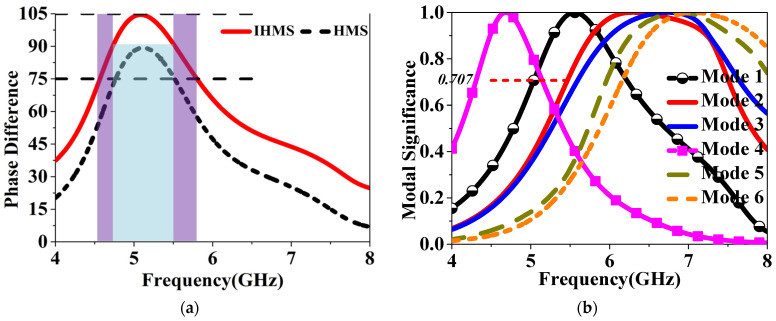
(**a**) CAs differences between HMS and IHMS and (**b**) MSs of the first six modes of the IHMS.

**Figure 19 sensors-24-03510-f019:**
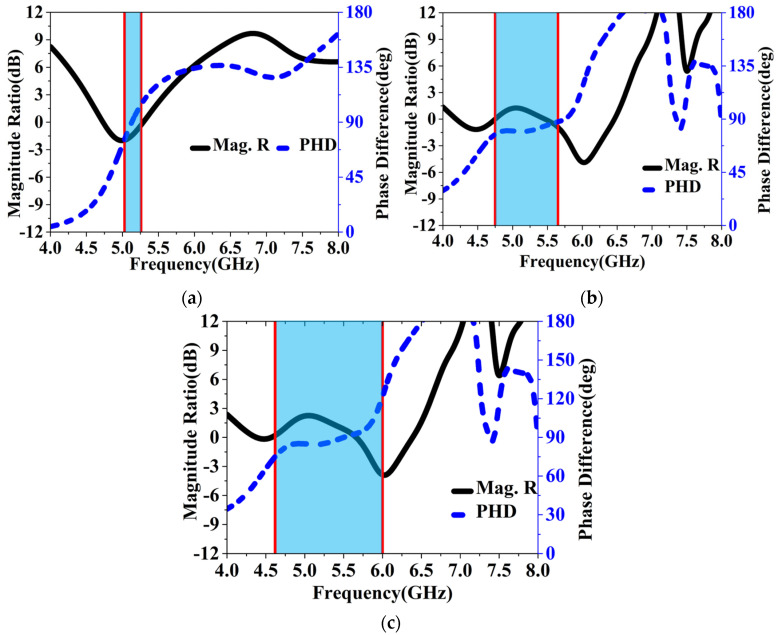
Simulated E-field magnitude ratio and phase difference in far–field for (**a**) RMS, (**b**) HMS, and (**c**) IHMS.

**Figure 20 sensors-24-03510-f020:**
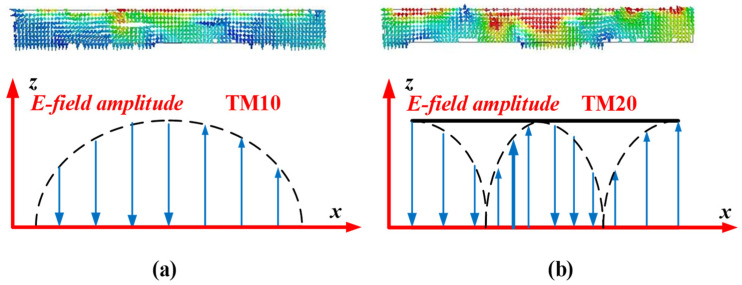
Simulated E-field distributions along the center of the IHMSA for (**a**) TM10 and (**b**) TM20 at 5.2 GHz and 6.2 GHz, respectively, as well as a schematic of the operational mechanism.

**Figure 21 sensors-24-03510-f021:**
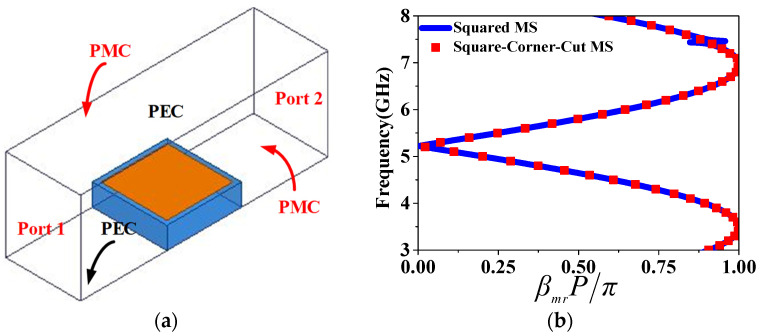
(**a**) Transmission-line model in HFSS.15 and (**b**) dispersion diagrams of the square MS and square-corner-cut MS unit cell.

**Figure 22 sensors-24-03510-f022:**
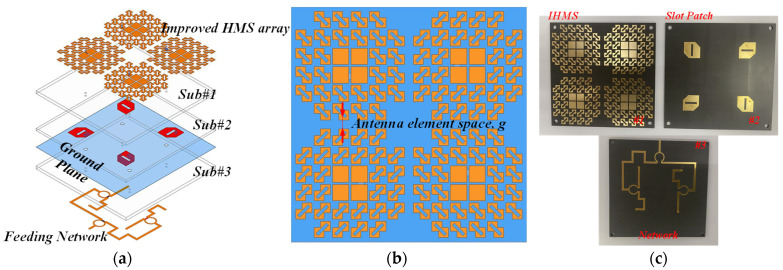
The sketch of the improved HMS–based slot patch antenna array: (**a**) 3D view and (**b**) top view, and (**c**) photos of the fabricated array antenna.

**Figure 23 sensors-24-03510-f023:**
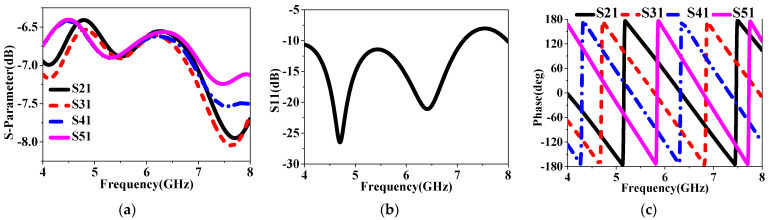
The behaviors of the feeding network: (**a**) S parameter, (**b**) S11, and (**c**) phase.

**Figure 24 sensors-24-03510-f024:**
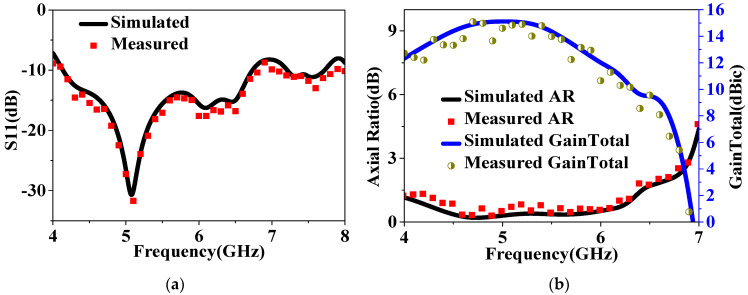
Simulated and measured results of the proposed IHMS–based patch array antenna: (**a**) S11 and (**b**) axial ratio and gain total.

**Figure 25 sensors-24-03510-f025:**
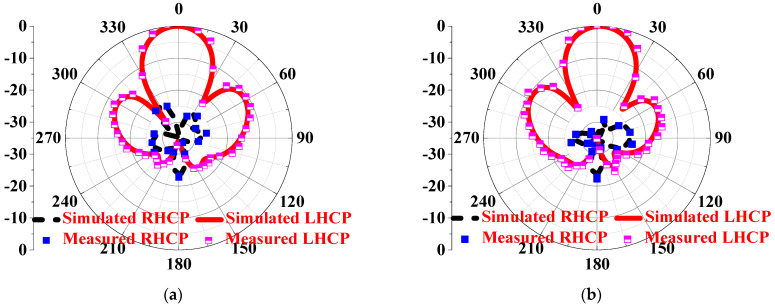
The normalization simulated and measured radiation patterns at 4.7 GHz in the (**a**) xoz plane and (**b**) yoz plane, respectively.

**Figure 26 sensors-24-03510-f026:**
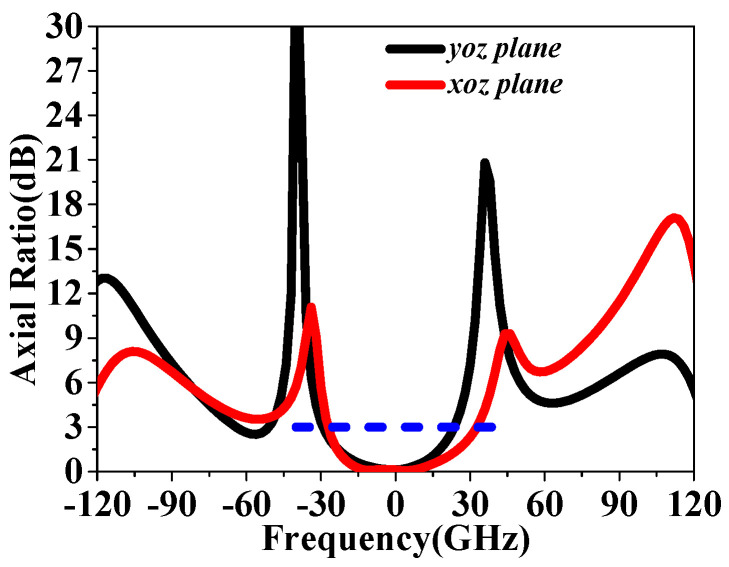
Simulated AR values versus the angle at 4.7 GHz in *xoz* and *yoz* plane, respectively, at 4.7 GHz.

**Table 1 sensors-24-03510-t001:** Optimal dimensions of proposed MS-based antenna.

Dimension	Size (mm)	Dimension	Size (mm)
*h_a_*	3.175	*L*	55
*W_s_*	2	*C*	5.3
*L_s_*	12	*W*	16.25
*g*	1	*L_px_*	7.625
*W_sy_*	1.5	*L_sx_*	7
*Cs*	3		

**Table 2 sensors-24-03510-t002:** Comparisons of radiated performances with the proposed RHCP IHMS-based antenna.

Types of Antennas	Structure	S11 (GHz)	3 dB AR (GHz)	Peak Gain (dBic)
LHCP ANTENNA	just 90° rotated IHMS-based	4.25–8.48	4.62–5.46	9.4
RHCP ANTENNA	IHMS-based	4.1–7.1	4.4–6.2	9.94

**Table 3 sensors-24-03510-t003:** Performance comparison of the recently reported CP antennas.

Ref.	Source Antenna Type	MS Type	Center Frequency (GHz)	Size (*λ*_0_^3^)	3 dB AR BW (GHz)	Peak Gain (dBic)	3 dB AR Angle Range	Operating Bandwidth (GHz)
Proposed	slot patch	Hybrid	5.175	1.79 × 1.79 × 0.09	4–6.35 (45.4%)	15.1	−30° to 30°	4–6.35 (45.4%)
[18]	slot patch	Hybrid	8.45	2 × 2 × 0.08	7–9.78 (33.13%)	13.17	−8° to 10°	6.05–10.04 (49.6%)
[19]	Slot Patch	Nonuniform	1.75	0.67 × 0.67 × 0.06	1.3–2.1 (47%)	8.7	−21° to 18°	1.4–2.1 (40%)
[20]	patch	uniform	10.2	2.6 × 2.63 × 0.36	9.8–10.2 (4%)	17.8	−15° to 15°	9.78–10.26 (4.79%)
[17]	Oblique slot patch	Nonuniform	2.145	2.0 × 2.0 × 0.68	2–2.25 (11.7%)	7.1	−28° to 28°	2–2.29 (13.6%)
[25]	L-shaped slot Patch	Uniform	6.85	2.0 × 2.0 × 0.88	1.36 × 1.36 × 0.08	10.5	−12° to 15°	5.2–8.5 (48.2%)
[26]	slot patch	Uniform	2.485	0.99 × 0.99 × 0.05	2.38–2.56 (7.3%)	10.01	-	2.27–2.7 (17.6%)
[27]	slot patch	Uniform	3.42	0.58 × 0.58 × 0.043	3.33–3.63 (8.5%)	6.57	-	3.02–3.82
[28]	slot patch	Uniform	5.32	0.65 × 0.65 × 0.06	5.18–6.19 (17.77%)	6.3	-	4.28–6.37 (39.25%)
[29]	slot patch	Nonuniform	5.6	0.56 × 0.56 × 0.076	(4.9–5.6) 12.78%	13.4	-	(5.0–6.3) 21.52

Note: “-” represents null.

## Data Availability

Dataset available on request from the authors.

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
