# Peer review of "Wideband Circularly Polarization and High-Gain of a Slot Patch Array Antenna Realized by a Hybrid Metasurface"

_sensors, 2024, doi:10.3390/s24113510_

Round 1
Reviewer 1 Report
Comments and Suggestions for Authors
This study proposes a novel patch antenna design for wideband circular polarization and high gain. It uses a hybrid metasurface (HMS) with two layers: a receiver layer to capture the linearly polarized wave and a conversion layer to transform it into a circularly polarized wave. Analysis and simulations confirm the wideband performance. The design is further improved and implemented as a 2x2 array antenna achieving excellent measured results.
1- The authors' decision to use distinct transmitter and reception stages in the HMS design requires further justification.
2-For easy comparison, think about including a table that summarizes the important performance parameters (gain, bandwidth, AR, etc.) for both LHCP and RHCP configurations.
3- some picture legends are not readable Fig 16 b X-axis Fig 20
Comments on the Quality of English Language
The manuscript exhibits good overall grammar with clear and concise language suitable for an academic publication.
Author Response
Reviewer #1:
This study proposes a novel patch antenna design for wideband circular polarization and high gain. It uses a hybrid metasurface (HMS) with two layers: a receiver layer to capture the linearly polarized wave and a conversion layer to transform it into a circularly polarized wave. Analysis and simulations confirm the wideband performance. The design is further improved and implemented as a 2×2 array antenna achieving excellent measured results.
Author response: Thank you for your positive judgment on this work, which allow us have an intention to explore this technique for exploration of hybrid MS-based wideband CP antenna arrays with enhanced AR bandwidth and realized gain with the low profile property in the future. We really appreciate your constructive comments and suggestions, which will help us a lot in improving the quality of our manuscript.
(1) The authors’ decision to use distinct transmitter and reception stages in the HMS design requires further justification.
Author response: Thank you for your comments. In this paper, we have analyzed the performance of the receiver MS (RMS) in comparison with an HMS-based antenna and a conventional MS that shares the same unit cell structure as the RMS, as illustrated in Fig.7. The design process was further validated using CMA. The modal behaviors of PCMS, as shown in Fig.10, Fig.11, Fig.12, Fig.13, and Fig.14, can be effectively integrated with those of RMS, resulting in exceptional properties. Additionally, the E-field analysis demonstrates that the HMS-based antenna exhibits superior CP radiated properties, as depicted in Fig.15. If the explanation and justification for the utilization of HMS is unclear, we will make every effort to incorporate your valuable suggestions. Thank you for your comments once again.
(2) For easy comparison, think about including a table that summarizes the important performance parameters (gain, bandwidth, AR, etc.) for both LHCP and RHCP configurations.
Author response: Thanks for your valuable comments and suggestions, which will help us a lot in improving quality of our manuscript. Based on your comments, the important performance metrics have been included in the revised manuscript in Table I.
(3) Some picture legends are not readable Fig 16 b X-axis Fig 20
Author response: Thanks for your comments. We apologize for our carelessness during the writing process, which resulted in Figures 16-20 being unreadable. We have made the necessary modifications to improve their clarity, and the details have been highlighted in red font in the revised manuscript.

Reviewer 2 Report
Comments and Suggestions for Authors
1. Is there any distance between Metasurface and Patch Antenna? If yes, provide parametric study of changing distance betwen Metasurface and antenna; explain how the paramters of the antenna such as, S-Paramters, AR and gain are affecting.
2. In Table 2,
- write the make a column for %BW for better understand the comparison.
- Size of the antenna is with metasurface or only source antenna?
- 3dB AR angle range column is mostly "null". Select those papers where the required information is available.
- It will be good if Table 2 is further explained by adding more columns, discussing other paramters of the proposed design. For example, Source antenna type, center frequency etc.
3. Figure 18,
-why the source antenna array is not symmetrical? 2 x 2 array patch elements are same diagonally, however different when opposite to each other? What is the advantage of this methodology? Explain.
- Each element is feeded individually or with feeding network? Explain the behaviour of feeding network alone and in correspondence to the patch antenna elements with the help of S-parameters, AR and gain.
4. Fabricated model is not present in the manuscript. Add the pictures of final realized structure.
5. What are the real practical applications of this proposed structure? Why the operating frequency range is selected 4-6.35GHz. Explain?
6. Describe clearly why the hybrid Metasurface is better compared to conventional Metasurface? For example, compare the proposed structures with the following,
- https://ieeexplore.ieee.org/abstract/document/8657964?casa_token=avQ6QW2CAmgAAAAA:DlGVh_uhA7CDbsy_lyz-xYQdZ5o46e9GNy8YLfal0fkN1HMykZqBmi_1hfSaPx2aMJF_0PBXT20
- https://ieeexplore.ieee.org/abstract/document/6529103?casa_token=LFem7Hh84XgAAAAA:4BxA7360iollgWZEHnJbtiyYxvtyosfZ6O09srJK4V8vNuSIRZEk0aJBjPZ8gKFaFzdc8lNIcvY
- https://www.sciencedirect.com/science/article/pii/S1434841123004983
- https://www.mdpi.com/1996-1944/13/5/1164
7. Overall structure of the paper is confusing. Rewrite the paper in a good format to understand the proposed work clearly.
Comments on the Quality of English LanguageExtensive improvement is required.
Author Response
Reviewer #2:
- Is there any distance between Metasurface and Patch Antenna? If yes, provide parametric study of changing distance betwen Metasurface and antenna; explain how the parameters of the antenna such as, S-Parameters, AR and gain are affecting.
Author response: Thanks for your comments. The substrate #1 is sandwiched between the MS and the slot patch antenna, with a height denoted as hm. In response to your comments, an investigation of the effect of hm on S11, AR, and gain has been conducted in the revised manuscript, which is highlighted in red font.
- in Table 2,
- write the make a column for %BW for better understands the comparison.
Author response: Thank you for your comments. We have included the %BW in the revised manuscript, which is highlighted in red font.
- Size of the antenna is with metasurface or only source antenna?
Author response: Thank you for your comments. The size of the MS is included in the dimensions of the antenna.
-3dB AR angle range column is mostly “null”. Select those papers where the required information is available.
Author response: Thank you for your comments. The properties of the newly added works, for which the properties are available, are included in the revised manuscript, highlighted in red font.
- It will be good if Table 2 is further explained by adding more columns, discussing other parameters of the proposed design. For example, Source antenna type, center frequency etc.
Author response: Thank you for your comments. According to your valuable suggestions, we have added more parameters for comparison.
- Figure 18, -why the source antenna array is not symmetrical? 2 × 2 array patch elements are same diagonally, however different when opposite to each other? What is the advantage of this methodology? Explain.
Author response: Thank you for your comments. In fact, the source antenna array is symmetrical; however, it is possible that the asymmetry results from the moving operations required for clear observation in a 3D view.
- Each element is feeded individually or with feeding network? Explain the behavior of feeding network alone and in correspondence to the patch antenna elements with the help of S-parameters, AR and gain.
Author response: Thank you for your valuable suggestions. The array antenna is fed with a network that has equal magnitude but a 90° phase difference. The behaviors of the network are investigated in the revised manuscript, which is highlighted in red font.
- Fabricated model is not present in the manuscript. Add the pictures of final realized structure.
Author response: Thank you for your helpful suggestions. The fabricated pictures have been included in the revised manuscript, which is highlighted in red font.
- What are the real practical applications of this proposed structure? Why the operating frequency range is selected 4-6.35GHz. Explain?
Author response: Thanks for your comments. The proposed antenna is suitable for C-band communication applications, including military and civilian communication, as well as point-to-point links. The operating frequency is determined based on the overall dimensions of the HMS antenna, which measures 55mm and is equivalent to two and a half wavelengths. In this study, we have introduced a design method to achieve enhanced bandwidth and improved gain. Therefore, when another frequency band is required, we can simply adjust the dimensions while also utilizing this design method.
- Describe clearly why the hybrid Metasurface is better compared to conventional Metasurface? For example, compare the proposed structures with the following,
Author response: Thank you for your comments. In this paper, we have analyzed the performance of the receiver MS (RMS) in comparison with an HMS-based antenna and a conventional MS that shares the same unit cell structure as the RMS, as illustrated in Fig.7. The design process was further validated using CMA. The modal behaviors of PCMS, as shown in Fig.10, Fig.11, Fig.12, Fig.13, and Fig.14, can be effectively integrated with those of RMS, resulting in exceptional properties. Additionally, the E-field analysis demonstrates that the HMS-based antenna exhibits superior CP radiated properties, as depicted in Fig.15. If the explanation and justification for the utilization of HMS is unclear, we will make every effort to incorporate your valuable suggestions.
Next, it can be observed from the following four works that all MSs were uniformly arranged in these designs and exhibited linear-to-circular polarization properties. In this particular design, a novel circular polarization design method was proposed in which receive and transmit stages were separated to reduce phase cancellation at specific frequencies, leading to improved properties.
Then, as demonstrated in Fig. 7, the simulated results confirm that the HMS design achieves significantly enhanced radiated performance compared to the uniformly arranged CMS. While the improvement in radiated properties may be attributed to the MS structure in this design, it represents a novel approach to enhancing performance, to the best of our knowledge.
Finally, Table III lists the comparisons with the following works to better validate our design.
-https://ieeexplore.ieee.org/abstract/document/8657964?casa_token=avQ6QW2CAmgAAAAA:DlGVh_uhA7CDbsy_lyz-xYQdZ5o46e9GNy8YLfal0fkN1HMykZqBmi_1hfSaPx2aMJF_0PBXT20
Author response: Thanks for your comments. The proposed antenna is suitable for C-band communication applications, including military and civilian communication, as well as point-to-point links.
-https://ieeexplore.ieee.org/abstract/document/6529103?casa_token=LFem7Hh84XgAAAAA:4BxA7360iollgWZEHnJbtiyYxvtyosfZ6O09srJK4V8vNuSIRZEk0aJBjPZ8gKFaFzdc8lNIcvY
- https://www.sciencedirect.com/science/article/pii/S1434841123004983
- https://www.mdpi.com/1996-1944/13/5/1164
- Overall structure of the paper is confusing. Rewrite the paper in a good format to understand the proposed work clearly.
Author response: Thank you for your continued suggestions. We have adjusted the writing structure to ensure clarity, and the details are highlighted in red font in the revised manuscript.
As indicated in the revised manuscript, it has been reorganized into the following seven parts: introduction, CP antenna element design process and operating mechanism, antennas analysis, radiating mechanism of the antenna, simulated and measured results, and conclusion, respectively. The logic circuit is designed based on structural evolution, followed by the working principles of the units, overall antenna performance comparison analysis, radiation mechanism, and finally simulation and testing results. The LHCP IMS-antenna design and analysis have been moved to Part 2 from the simulated part.

Round 2
Reviewer 2 Report
Comments and Suggestions for Authors
Authors answered all the questions.
Comments on the Quality of English LanguageRevised manuscript is in good form.